# Comprehensive analysis of differential expression profiles via transcriptome sequencing in SH-SY5Y cells infected with CV-A16

Yajie Hu[1,2☯], Zhen Yang[1,2☯], Shenglan Wang[1,2], Danxiong Sun[1,2], Mingmei Zhong[1,2], Mudong Wen[1,2], Jie Song[3]*, Yunhui Zhang[1,2]*

1 Department of Respiratory Medicine, The First People's Hospital of Yunnan province, Kunming, China,
2 The Affiliated Hospital of Kunming University of Science and Technology, Kunming, Yunnan, China,
3 Yunnan Key Laboratory of Vaccine Research and Development on Severe Infectious Diseases, Institute of Medical Biology, Chinese Academy of Medical Science and Peking Union Medical College, Kunming, China

☯ These authors contributed equally to this work.
* songjiekm@163.com (JS); zhangyh123kh@163.com (YZ)

**Data Availability Statement:** All relevant data are within the manuscript and its Supporting Information files. The sequencing data have been submitted to the Gene Expression Omnibus (GEO)

## Abstract

Coxsackievirus A16 (CV-A16) is one of the viruses that is most frequently associated with hand-foot-and-mouth disease (HFMD). Previous studies have shown that CV-A16 infections are mostly self-limiting, but in recent years, it has been gradually found that CV-A16 infections can also induce neurological complications and eventually cause death in children with HFMD. Moreover, no curative drugs or preventative vaccines have been developed for CV-A16 infection. Therefore, it is particularly important to investigate the mechanism of CV-A16 infection-induced neuropathy. In the current study, transcriptome sequencing technology was used to identify changes in the transcriptome of SH-SY5Y cells infected with CV-A16, which might hide the mechanism of CV-A16-induced neuropathology. The transcriptome profiling showed that 82,406,974, 108,652,260 and 97,753,565 clean reads were obtained in the Control, CV-A16-12 h and CV-A16-24 h groups, respectively. And it was further detected that a total of 136 and 161 differentially expressed genes in CV-A16-12 h and CV-A16-24 h groups, respectively, when compared with Control group. Then, to explore the mechanism of CV-A16 infection, we focused on the common differentially expressed genes at different time points of CV-A16 infection and found that there were 34 differentially expressed genes based on which clustering analysis and functional category enrichment analysis were performed. The results indicated that changes in oxidation levels were particularly evident in the GO term analysis, while only the "Gonadotropin-releasing hormone receptor pathway" was enriched in the KEGG pathway analysis, which might be closely related to the neurotoxicity caused by CV-A16 infection. Meanwhile, the ID2 closely related to nervous system has been demonstrated to be increased during CV-A16 infection. Additionally, the data on differentially expressed non-protein-coding genes of different types within the transcriptome sequencing results were analyzed, and it was speculated that these dysregulated non-protein-coding genes played a pivotal role in CV-A16 infection. Ultimately, qRT-PCR was utilized to validate the transcriptome sequencing findings, and the

database (www.ncbi.nlm.nih.gov/geo/) under the accession number GSM4593022 (Control group), GSM4593023 (CV-A16-12 h group) and GSM4593024 (CV-A16-24 h group).

**Funding:** This study is supported by Yunnan Applied Basic Research Projects (2019FB018 and 2018ZF006), Doctoral Research Fund of the First People's Hospital of Yunnan Province (KHBS-2020-013), National Natural Sciences Foundations of China (31700153), Fundamental Research Funds for the Central Universities and PUMC Youth Fund (3332019004), Medical Reserve Talents of Yunnan Province Health and Family Planning (H-2017034) and Top young talents of Yunnan province ten thousand talents plan (Jie Song). The funders had no role in the study design, data collection and analysis, decision to publish, or preparation of the manuscript.

**Competing interests:** The authors have declared that no competing interests exist.

results of qRT-PCR were in agreement with the transcriptome sequencing data. In conclusion, transcriptome profiling was carried out to analyze response of SH-SY5Y cells to CV-A16 infection. And our findings provide important information to elucidate the possible molecular mechanisms which were linked to the neuropathogenesis of CV-A16 infection.

# 1. Introduction

Hand, foot, and mouth disease (HFMD), a common childhood illness, is caused by a large array of enteroviruses, especially enterovirus 71 (EV-A71) and coxsackievirus A16 (CV-A16), and is characterized by fever, oral ulcers, and skin manifestations affecting the palms, soles, and buttocks [1, 2]. Usually, these clinical manifestations of HFMD spontaneously resolve, but sometimes, a handful of patients with serious complications may evolve into severe illness, such as aseptic meningitis, encephalitis, acute flaccid paralysis (AFP), and even fatal myocarditis and pneumonia [3]. Moreover, a large number of severe and fatal cases of HFMD have been found to be closely associated with EV-A71 infection [4, 5]. Thus, previous studies have mainly focused on EV-A71 but not CV-A16. Fortunately, the first inactivated EV-A71 vaccine was successfully developed by the Institute of Medical Biology, Chinese Academy of Medical Science (CAMS), and has been licensed by the China Food and Drug Administration (CFDA) at the end of 2015 [6]. Therefore, this vaccine was undoubtedly believed to bring good news to millions of children [7]. However, it only protects against a fraction of HFMD cases caused by EV-A71 and cannot effectively control the HFMD epidemics induced by other enteroviruses, including CV-A16 [8]. Furthermore, in recent years, a dramatic increase in HFMD cases caused by CV-A16 has been reported [9]. Additionally, numerous studies have revealed that although the majority of HFMD patients with CV-A16 infection present only mild symptoms, patients with CV-A16 infection may in some cases develop severe central nervous system (CNS) complications and even die [10]. Hence, these findings suggested that similar to EV-A71, CV-A16 is a neurotropic virus and is responsible for severe neurological outcomes [10].

According to previous EV-A71-associated studies, it has been reported that neuronophagia and neuron necrosis were found in the brainstem and spinal cord of patients in most fatal cases accompanied by CNS complications [5]; meanwhile, viral antigens and RNA of EV-A71 were also detected in neurons [4]. Furthermore, a number of neurotropic viruses, such as Borna disease virus (BDV), Japanese encephalitis virus (JEV), and cytomegalovirus (CMV), could cause damage to the CNS through disrupting the differentiation, proliferation and lifespan of neurons [11]. Thus, based on the underlying neurotropic features of CV-A16, it was speculated that CV-A16 could productively infect human neurons. In this study, the human neuroblastoma cell line SH-SY5Y was adopted to construct a CV-A16-infected cellular model and then the transcriptome profiles in CV-A16-infected SH-SY5Y cells were dissected to preliminarily reveal the underlying neuropathogenesis of CV-A16. These results might be provide new strategies for developing effective and specific medication and vaccines to better control CV-A16 infections in epidemic areas.

# 2. Materials and methods

## 2.1. Cell cultivation, virus inoculation sample collection and ethics statement

The human neuroblastoma cell line SH-SY5Y, purchased from Jennino Biological Technology, was propagated in Dulbecco's modified Eagle's medium (DMEM) (Corning, USA) with the

addition of 10% fetal bovine serum (FBS; Gibco, USA), 2 mmol/L glutamine, 100 units/mL penicillin and 100 μg/mL streptomycin (Gibco, USA) at 37°C in a humidified 5% $CO_2$/95% air atmosphere until the cell lines reached 90% confluence. Then, the cells were washed twice with phosphate-buffered saline (PBS) and continued to be passaged with 0.25% trypsin (Sigma, USA) and grown in fresh complete culture medium.

The CV-A16-G20 strain (sub-genotype B, GenBank: JN590244.1), isolated from an HFMD patient in Guangxi, China, in 2010, was used in this study. SH-SY5Y cells were seeded into 6-well plates at a density of $5 \times 10^5$ cells/well and infected with CV-A16 at a multiplicity of infection (MOI) of 1. Subsequently, the infected cells were harvested at 0, 12 and 24 h post infection (hpi). Moreover, cells treated with CV-A16 for 0 hpi were defined as control group.

In this study, all of the following experiments we conducted were performed at the cellular level, and no experiments related to animals and humans were done. Therefore, this study does not involve ethical issues. We hereby declared.

## 2.2. Examination of CV-A16 proliferation kinetics

The detection of virus proliferation kinetics could utilize the determination of virus titer. SH-SY5Y cells were plated into 6-well plates at a density of $5 \times 10^5$ cells/well and infected with CV-A16 at a multiplicity of infection (MOI) of 1. Subsequently, the infected cells were harvested at 6, 12, 24 and 36 h post infection (hpi). In order to examination of the virus titer used plaque assay, Vero cells were grown to a monolayer on 6-well plates and then inoculated with serial 10-fold dilutions of the above samples (1 ml per well). After 3 h of incubation to allow virus attachment, the wells were gently washed with PBS, covered with media containing 1% agarose and placed into a 37°C $CO_2$ incubator for 48 h. Next, the cells were fixed with 2 ml of 4% paraformaldehyde (PFA) (Solarbio, China) and incubated for 30 min at room temperature, and the 1% agarose was removed.The monolayer of cells was stained with a crystal violet staining solution for 15 min, and washed with ddH$_2$O. Finally, visible plaques were counted by the naked eye and the plaque-forming units (pfu/ml) were calculated with the virus titer formula, where virus titer equals the number of plaques × (1 ml) × dilution factor. In addition, in order to observe the cytopathic effect (CPE) of SH-SY5Y cells with CV-A16 infection, we capture the photos of SH-SY5Y cells infected with CV-A16 and PBS (Control) using LEICA DMi8 (S1 Fig).

## 2.3. RNA extraction, library preparation and transcriptome sequencing

Total RNA was extracted from three independent experimental replicates of each group which subsequently pooled together using the TRIzol solution (Invitrogen, USA) in accordance with the manufacturer's recommendations. The isolated RNA was treated with DNase I (Thermo, USA) at 37°C for 1 h to ensure that genomic DNA was eliminated from the samples. RNA quality was detected using GeneGreen-stained 1% non-denaturing agarose gel electrophoresis, the RNA concentration was checked by analyzing the ratios A260/280 using a NanoDrop2000 Spectrophotometer (Thermo Scientific, USA) and RNA integrity was assessed with the Agilent 2200 TapeStation analysis (Agilent Technologies, USA). Finally, only high quality total RNA (i.e., RIN > 7) was applied as input material for the subsequent library construction.

RNA sequencing (RNA-seq) libraries were constructed using the NEBNext® Ultra™ RNA Library Prep Kit (NEB, USA) following the manufacturer's protocol. In brief, 10 μg of total RNA that passed RNA quality control (QC) measures was purified to obtain poly-A-containing mRNA molecules using poly-T oligo-attached magnetic beads. After purification, the mRNA is fragmented into small pieces using divalent cations under elevated temperature in the NEB Next First Strand Synthesis Reaction Buffer 5 (×). The cleaved short RNA fragments

were reverse transcribed into first strand cDNA with ProtoScript II reverse transcriptase (NEB, USA) by a random hexamer primer. Then, the second strand cDNA synthesis was further obtained using DNA polymerase I, dNTPs, and RNase H. Following an end repair process and the addition of a single 'A' base, the cDNA fragments were ligated to sequencing adapters and amplified by polymerase chain reaction (PCR, 15-cycle) to create the final cDNA library. Eventually, the libraries were sequenced on an Illumina Sequencing System (HiSeq2000) using paired-end technology according to the manufacturer's standard workflow. The sequencing data were deposited in the National Center for Biotechnology Information's Gene Expression Omnibus (GEO) database (www.ncbi.nlm.nih.gov/geo/) under accession number GSM4593022 (Control group), GSM4593023 (CV-A16-12 h group) and GSM4593024 (CV-A16-24 h group).

### 2.4. Analysis of RNA-seq data

**2.4.1. Data filtering and mapping reads.** The raw paired-end reads were obtained from each sample. After quality assessment with the Fast QC package (http://www.bioinformatics. babraham.ac.uk/projects/fastqc/), the "dirty" reads that contained numerous interspersed N's in their sequences, or had relatively short reads (<17 bp), were removed according to the default parameters for subsequent analysis. Next, all clean tags were mapped to the GRCh38 human reference genome using HISAT2 software.

**2.4.2. Gene structure and reads distribution on the chromosome analysis.** Gene structure mainly involves four types of regions, namely: the coding region (e.g., exons, introns, etc.), the leader region (5'-terminal non-coding region), the tail region (3'-terminal non-coding region) and the regulation regions (e.g., promoters, enhancers, etc.). The analysis of gene structure may be useful for identifying the differences between normal and viral infections. In addition, mapping the picture of reads distribution on the chromosome can provide a general understanding of the sequencing results of the transcriptome, thereby assisting in determining the reads coverage of the interesting fragments on the chromosome. Moreover, under normal circumstances, the longer the entire chromosome is, the more total reads it will be located internally.

**2.4.3. Differential expressed genes analysis.** The quantified number of mapped reads was normalized as fragments per kilobase of transcripts per million fragments mapped (FPKM) value. The false discovery rate (FDR) was used to predict the *P* value threshold for statistical analysis. Differentially expressed genes infected and non-infected groups, were identified at combined cut offs of as FDR < 0.05 and FPKM value ≥ 1.5 using Cufflinks software (version 2.1.1). Subsequently, these differentially expressed genes were further used for the global classification, including transfer RNA (tRNA), small nuclear RNA (snRNA), Protein-coding, Precursor_miRNA, non-coding RNA (ncRNA) and microRNA (miRNA).

**2.4.4. Clustering analysis.** Firstly, a Venn diagram was used to find out the concordant (up-regulated in both or down-regulated in both) differentially expressed genes between the CV-A16-12h group and the CV-A16-24h group. Then, in order to further determine the specifcity of the co-expression genes between CV-A16-infected groups and their control groups, unsupervised hierarchical clustering analysis was utilized for the clustering of distinct sample groups. Unsupervised clustering was performed on the normalized and log2-transformed differentially expressed genes data by calculating Pearson's centered correlation coefficient followed by average linkage analysis. Ultimately, the resulting output was used to generate the associated heatmap and clustering dendrogram with version 3.0 of Cluster software.

**2.4.5. Functional category enrichment analysis.** To comprehensively investigate the potential roles of the indicated differentially expressed genes, Gene ontology (GO) enrichment

analysis and Kyoto encyclopedia of genes and genomes (KEGG) pathway enrichment analysis of these genes were conducted by using the Database for Annotation, Visualization, and Integrated Discovery (DAVID, https://david.ncifcrf.gov/) online system. Moreover, GO analysis mainly included biological process (BP), cellular component (CC) and molecular function (MF) terms. Additionally, the KEGG pathway analysis also enabled the annotations of metabolic pathways and revealed the interactions among the significantly enriched pathways.

**2.4.6. Screening for differentially expressed non-coding protein genes.** Previously we focused on differentially expressed protein-coding genes. Next, we will continue to find non-coding protein genes that are differentially expressed during CV-A16 infection, especially the top 10 with significant changes.

## 2.5. Western blotting examination

WB analysis was performed according to a standard method. Briefly, total proteins from cell samples (including Control, CV-A16-12 h and CV-A16-24 h groups) were extracted using radioimmunoprecipitation assay (RIPA) buffer containing 50 mM Tris-HCl (pH 7.5), 150 mM NaCl, 0.1% Nonidet P-40, and a mixture of protease inhibitors, and the concentration of the proteins were detected by a BCA protein assay kit (Beyotime, China). A total of 30 μg of protein lysates from different groups was resolved by 8~10% sodium dodecyl sulfate-polyacrylamide gels for electrophoresis (SDS-PAGE) under 60 V for 2 h, and transferred to polyvinylidene difluoride (PVDF; Millipore, USA) membranes at a constant current of 200 mA for 60 min. After blocking with 5% skimmed milk at room temperature for 1 h, the PVDF membranes were incubated with primary antibodies against ID2 (1:1000 dilution; Abcam, USA) and GAPDH (1:500 dilution; Boster, China) overnight at 4˚C. Following thrice-washing with 1× TBST, the membranes were further incubated with the corresponding secondary antibodies (including goat anti-mouse IgG and goat anti-rabbit IgG, 1:12000 dilution; Boster, China) for 1 hour at 37˚C. Membranes were extensively washed with TBST three times again and finally visualized using the ECLWestern blot detection kit (Amersham, USA) and exposed to X-ray films (Kodak, Japan). Relative expression levels of each protein were normalized to the endogenous control, GAPDH, using ImageJ 1.8.0 software.

## 2.6. Verification by qRT-PCR

The validity of the RNA-Seq data was verified by qRT-PCR. Six genes with differential expression at were randomly chosen, including an upregulated gene (i.e., TXNIP), downregulated genes (i.e., CDK6, CHGA, FOSB and ZNF704) and a gene showing reverse regulation at different times (i.e., COX2). To ensure the accuracy of the validation, we collected samples that were treated the same as the samples on which the transcriptome analysis was performed.

Firstly, TRIzol solution (Invitrogen, USA) was used to isolate the total RNA from samples as described above. The quality and concentrations of total RNA were determined by a Bio-Photometer and the integrity of total RNA was assessed by agarose-formaldehyde gel electrophoresisthe. Then, 1 μg of total RNA from each sample was reverse transcribed into cDNA using the PrimeScript RT reagent kit with gDNA Eraser (Takara, Japan). Ultimately, PCR was employed using 2 μl diluted cDNA products were added to 12.5 μl SYBR Premix Ex Taq II (Takara, Japan), 0.5 μl forward and reverse primers (10 μM) and 9.5 μl nuclease-free water in a final volume of 25 μl on an ABI Prism 7900 Sequence Detection System (Applied Biosystems, USA) with a thermocycling protocol of 50˚C for 2 min, followed by 40 cycles of denaturation at 95˚C for 15 s and 60˚C for 1 min and annealing/elongation step at 60˚C for 30 s. The expressions of the above genes were normalized to GAPDH (an endogenous control) and calculated with the $2^{-\Delta\Delta Ct}$ method. All qRT-PCR experiments were repeated at least 3 times and the

primer sequences were designed and synthesized by GenePharma (Shanghai, China) (as shown in S1 Table).

## 2.7. Statistical analysis

For the sequencing data, the raw reads obtained from each library were normalized to FPKM values. For qRT-PCR, the data are presented as the mean ± standard error of the mean (SEM) according to statistical analysis with SPSS 18.0 software (IBM SPSS, USA). All experiments were repeated at least three times, and $P < 0.05$ indicated a statistically significant difference.

## 3. Results

### 3.1. Summary of RNA-seq data

RNA-seq, which is a genome-wide analytical technology that serves as the basis and starting point for the study of gene function and regulatory networks, has been extensively utilized to analyze the transcriptomes of various infectious diseases. In this study, we adopted RNA-seq to investigate the host-pathogens interaction in SH-SY5Y cells following CV-A16 infection. The results showed that 87,327,240, 110,800,518 and 104,446,596 raw reads were generated for the control, CV-A16-12 h and CV-A16-24 h groups, respectively (Table 1). After stringent data filtering, the remaining clean reads were obtained from the 3 groups, which consisted of 82,406,974, 108,652,260 and 97,753,565 clean reads, respectively. Therefore, it was calculated that the proportion of clean reads in the 3 samples was greater than 90%, suggesting a set of reliable sequencing data. Additionally, the quality scores across all bases and the GC content were also analyzed. As illustrated in S2 Fig, it was found that the quality scores across all bases were > 30, and the average of GC content was very close to 43% (a theoretical GC distribution), which further indicated the high quality of the sequencing of these samples and a satisfactory level for the subsequent study. Finally, the clean reads were unambiguously mapped against the GRCh38 human reference genome using HISAT2 software, and the unique mapped reads and repeat mapped reads were also identified. In addition to comparative analysis of reference genes in clean reads, we also performed gene structure and reads distribution on the chromosome analysis (Seen in S3 Fig). It was observed that there were no obvious differences between the Control group and the CV-A16-12 h group in gene structure, but the reads number of gene structure analysis was dramatically decreased in the CV-A16-24 h group when compared to the Control group and the CV-A16-12 h group. Meanwhile, it also found that the reads distributed on chromosome 21 were notably increased in the CV-A16-12 h group as compared with the Control group and the CV-A16-24 h group.

### 3.2. Modulation of the transcriptome profile in SH-SY5Y cells in response to CV-A16 infection

The comparison of the CV-A16-12 h and Control groups identified 136 genes (72 upregulated and 64 downregulated) that were significantly differentially expressed ($\geq \pm 1.5$ fold change between the two groups along with an FDR < 0.05) (Fig 1A). However, comparison of the

**Table 1. Basic characteristics of transcriptome sequencing.**

| Samples | Total reads | Clean reads | Mapping reads | Unique Mapped | Repeat Mapped |
|---|---|---|---|---|---|
| Control | 87,327,240 | 82,406,974 | 79,305,649 | 72,037,353 | 7,268,296 |
| CV-A16-12h | 110,800,518 | 108,652,260 | 101,643,271 | 89,012,108 | 12,631,163 |
| CV-A16-24h | 104,446,596 | 97,753,565 | 49,918,448 | 45,202,936 | 4,715,512 |

CV-A16-24 h and Control groups identified 161 genes (74 upregulated and 87 downregulated) that were significantly differentially expressed ($\geq \pm 1.5$ fold change between the two groups along with an FDR $< 0.05$) (Fig 1B). Moreover, the differentially expressed genes in both CV-A16-infected groups were chiefly consisted of tRNA, snoRNA, RNase_P_RNA, RNase_MRP_RNA, pseudo, Protein-coding, Precursor-miRNA, ncRNA, miscRNA, miRNA, and Unknown genes. Besides, there were differentially expressed telomerase_RNA and snRNA only in CV-A16-24 h group. Additionally, the detailed number of up-regulated and down-regulated genes in these different gene types is shown in S2 Table.

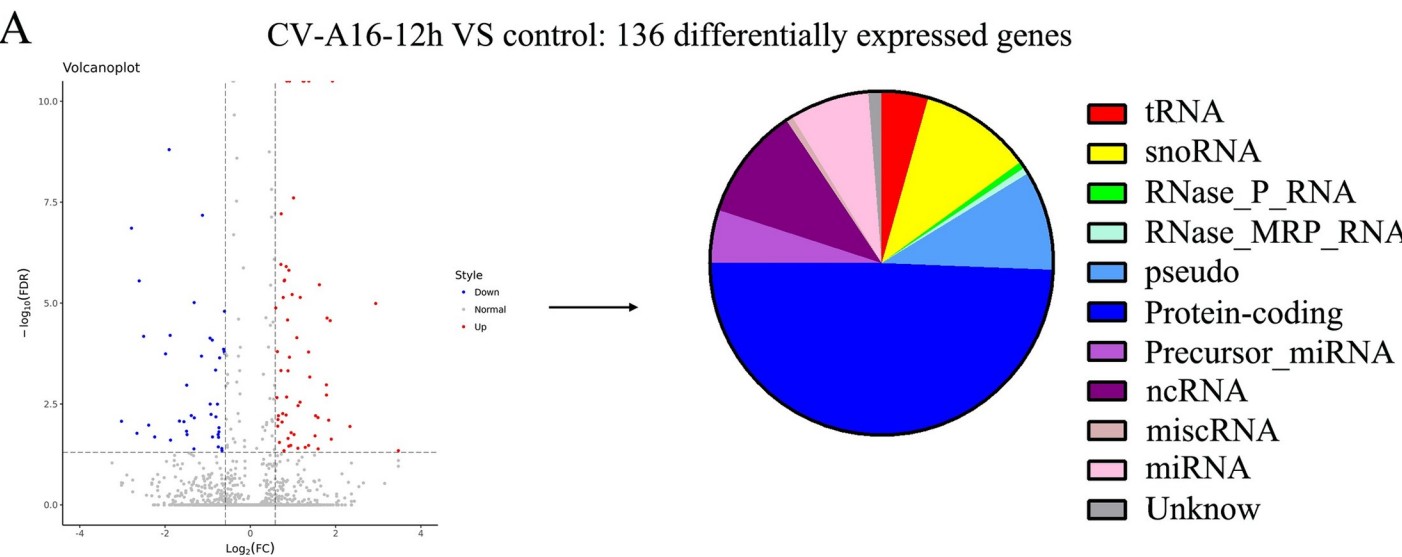

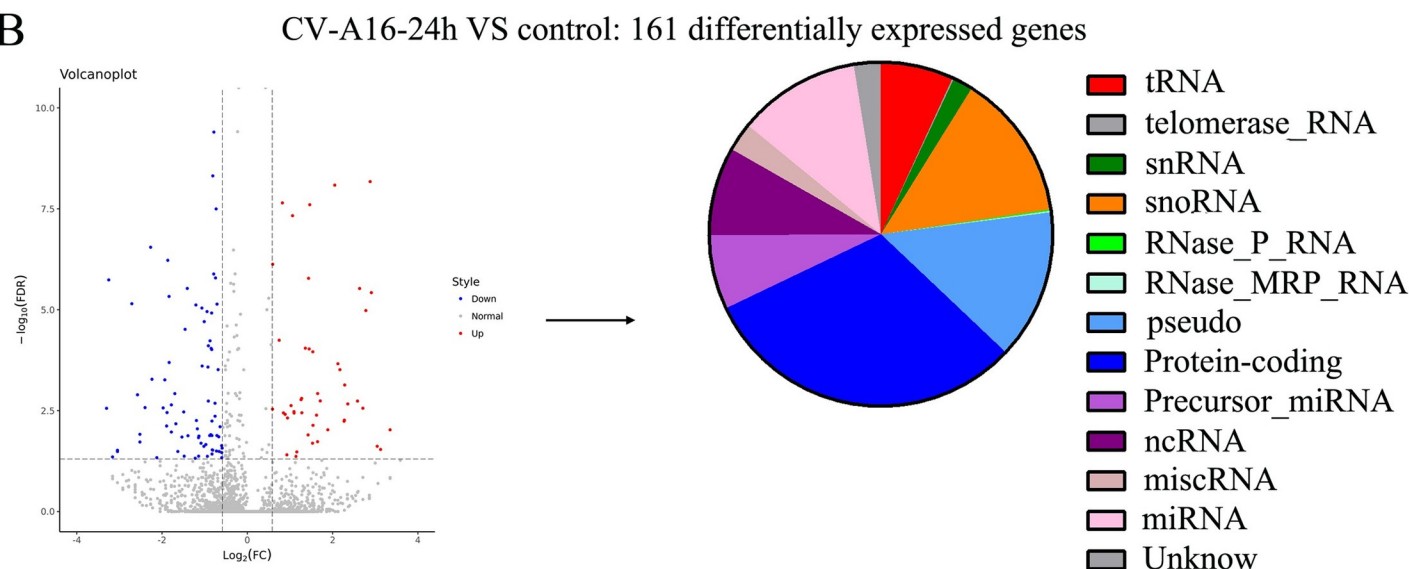

**Fig 1.** Differential expression of genes between the CV-A16-12 h group (A)/CV-A16-24 h group (B) and the control. Red dots represent genes that were upregulated in the CV-A16 infection groups compared to the control group, whereas blue dots represent genes that were downregulated in the CV-A16 infection groups compared to the control group. The pie chart shows the gene types of these differentially expressed genes during CV-A16 infection at different time points.

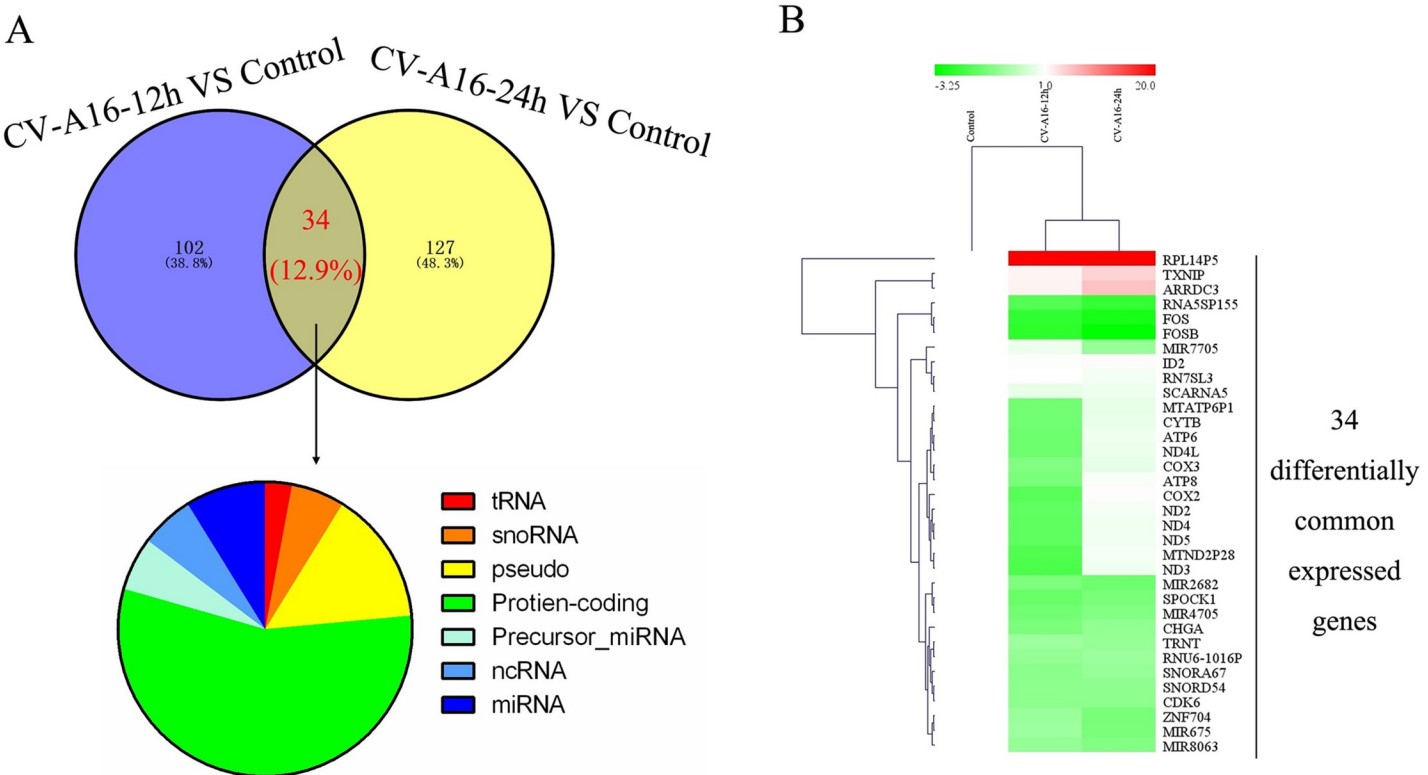

**Fig 2.** (A) Venn diagram analysis showing a total of 34 differentially expressed genes under different conditions. (B) Heat-map showing the increased (red) or decreased (green) expression trends of common differentially expressed genes during CV-A16 infection.

The Venn diagram of the differentially expressed genes illustrated that the number of CV-A16-12 h-specific, CV-A16-24 h-specific, and co-expressed genes were 102, 127 and 34, respectively (Fig 2A). To analyze the differences and similarities among the CV-A16-responsive transcriptomes, a hierarchical clustering approach was utilized to depict the 34 common differentially expressed transcripts. The hierarchical heatmap provided a clear visual summary of the dynamic changes in the transcriptional response to CV-A16 at two time points reflecting the pattern of expressional changes among the 34 differentially expressed transcripts (Fig 2B). Meanwhile, it was also observed that the CV-A16-12 h sample and the CV-A16-24 h sample were clustered together and separated from the control sample, which suggested that the host transcriptome responses in SH-SY5Y cells underwent significant changes during the process of CV-A16 infection.

### 3.3. Functional analysis of differentially expressed genes

To gain a better understanding of the gene functions and signaling pathways of CV-A16-infected-related differentially expressed genes, online GO and KEGG pathway enrichment analysis were conducted using DAVID. It was found that the 34 differentially expressed genes were obviously enriched in generation of precursor metabolites and energy, oxidative phosphorylation and respiratory electon transport chain at the "Biological process" level (Fig 3A), in oxidoreductase activity at the "Molecular function" level (Fig 3B) and in mitochondrial inner membrane at the "Cellular component" level (Fig 3C). Moreover, the enriched KEGG pathways of these 34 differentially expressed genes included only one pathway in Gonadotropin-

releasing hormone receptor pathway (Fig 4). Those differentially expressed genes related to "Oxidative phosphorylation" and "Gonadotropin-releasing hormone receptor pathway" were the focus of our attention and were listed in Tables 2 and 3.

### 3.4. WB detection

ID2 presented differentially expression was found in enriched "Gonadotropin-releasing hormone receptor pathway", and it is also a key gene associated with nervous system development. Thereby, it was selected to further examine it protein levels and it was observed that ID2 was obviously increased in SH-SY5Y cells in response to CV-A16 infection (Fig 5A).

### 3.5. Analysis of differentially expressed non-protein-coding genes

In the process of viral infection, in addition to protein-coding genes playing an important role, non-protein-coding genes may also be participated in the progression of the disease. Thus, at different time points of CV-A16 infection, we screened out differentially expressed non-coding protein genes that were common at the two infection time points (Illustrated in Table 4).

### 3.6. qRT-PCR validation

To assess the validity of the RNA-seq data, 6 differentially expressed genes were randomly selected for qRT-PCR analysis. The data indicated that there were no significant differences between qRT-PCR experiment results and the transcriptome sequencing results (Fig 5B).

## 4. Discussion

HFMD can be caused by any of several serotypes of human enteroviruses, most commonly EV-A71 and CV-A16 [2, 12]. CV-A16-associated HFMD was first reported in Canada in 1957, and CV-A16 infections are often asymptomatic and self-limiting diseases, but may also result in a diverse spectrum of clinical illnesses, varying from mild febrile illnesses to severe neurological disease and even death [6]. Although many researchers have made extensive efforts to understand CV-A16 infection in different human cell types [13–15], little is known about CV-A16 infection in SH-SY5Y cells. Moreover, as a neurotropic virus, it is particularly important to investigate the neuronal changes induced by CV-A16 infection. Therefore, in the current study, we employed high-throughput RNA-seq technology for the first time and obtained important information about the host-virus interaction in CV-A16-infected SH-SY5Y cells. SH-SY5Y cells, a subline of the parental line SK-N-SH, are humanderived, neuron-like cells and they express a number of human-specific proteins and protein isoforms [16]. SH-SY5Y cells are widely used in experimental neurological studies, including analysis of neuronal differentiation, metabolism, and function related to neurodegenerative and neuroadaptive processes, neurotoxicity, and neuroprotection, etc [17]. For example, neuron-like SH-SY5Y cells were applied to establish an *in vitro* cell model of Parkinson's disease [18]. Moreover, SH-SY5Y cells are also often used in *in vitro* research models of neurotropic viruses, including Herpes simplex virus 1 (HSV-1) [19], Enterovirus D68 (EV-D68) [20], Rabies virus (RABV) [21], Japanese encephalitis virus (JEV) [22], West Nile virus (WNV) [23], and so on. Nevertheless, CV-A16 is also a neurotropic virus, and its mechanism of causing pathological changes in the nervous system has not been elucidated.

Herein, we adopted SH-SY5Y cells as a cellular model to investigate the proliferation kinetics of CV-A16 on SH-SY5Y cells. And the result showed that CV-A16 could successfully proliferate on SH-SY5Y cells (S3 Fig), which suggested that SH-SY5Y cells could be used as an *in vitro* model of CV-A16 infection in neural cells to further explore the neuropathological

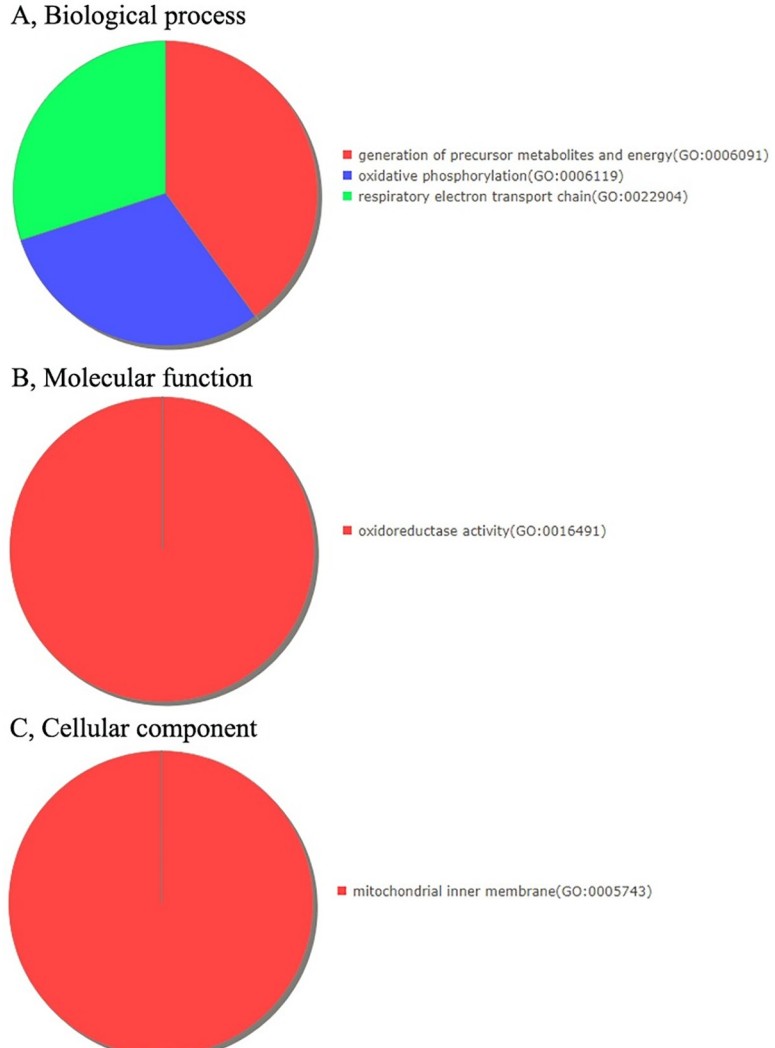

**Fig 3.** GO analysis indicating the enrichment of the dysregulated differentially expressed protein-coding genes in (A) Biological processes, (B) Molecular functions and (C) Cellular components.

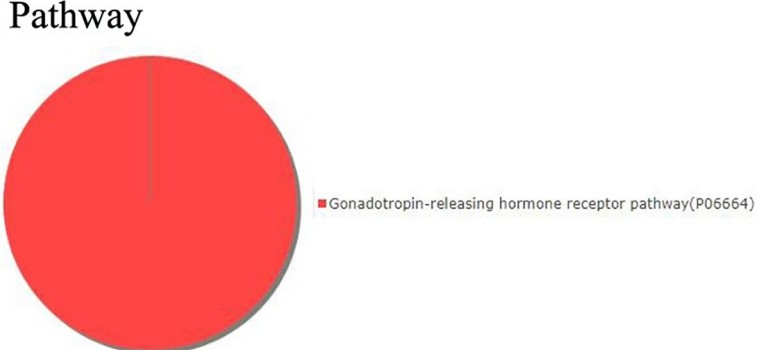

**Fig 4. The differentially expressed protein-coding genes were clustered and found to be enriched in the Gonadotropin-releasing hormone receptor pathway.**

**Table 2. Oxidative phosphorylation-associated differentially expressed genes in GO-BP analysis.**

| GO-BP | Gene Symbol | Gene Name | PANTHER Protein Class | CV-A16-12h | | CV-A16-24h | |
|---|---|---|---|---|---|---|---|
| | | | | Fold change | Log2FoldChange | Fold change | Log2FoldChange |
| Oxidative phosphorylation | COX2 | Cytochrome c oxidase subunit 2 | oxidoreductase | 0.29029806 | -1.784393164 | 2.45801905 | 1.297496097 |
| | COX3 | Cytochrome c oxidase subunit 3 | oxidase | 0.469888555 | -1.089609468 | 1.530889452 | 0.614370107 |
| | ND5 | NADH-ubiquinone oxidoreductase chain 5 | dehydrogenase | 0.321412715 | -1.637501092 | 1.697992889 | 0.763830417 |

mechanisms of CV-A16. Then, we have described the first global transcriptional profiles of SH-SY5Y cells infected with CV-A16 at different time points. The results revealed that 136 and 161 differentially expressed genes were identified in the CV-A16-12 h and CV-A16-24 h groups, respectively, via the screening of raw data. In order to find the commonalities in the process of CV-A16 infection, we used Venn analysis to obtain 34 genes with common differentially expression, which contained diverse gene types, such as tRNA, snoRNA, pseudo, Protein-coding, Precursor-miRNA, and miRNA. Moreover, the results of a hierarchical clustering showed that the CV-A16-12 h group and the CV-A16-24 h group were aggregated together, and meanwhile they clearly separated from the Control group. Subsequently, we selected the protein-coding genes among these differentially expressed genes for GO term and KEGG pathway analysis. It was found that 3 GO-BPs, 1 GO-MF, 1 GO-CC and 1 pathway were enriched. Among GOs analysis, the terms of "oxidative phosphorylation", "oxidoreductase activity" and "mitochondrial inner membrane" all involved oxidation levels, which implied that CV-A16 infection-induced changes in oxidation levels might be directly participated in the pathogenesis of CV-A16 infection. In fact, emerging evidence has demonstrated that virus-induced oxidative stress plays an important role in the regulation of the host immune system and contributes to several aspects of viral disease [24]. For example, Mayaro virus (MAYV) triggers significant oxidative stress in infected HepG2 cells and J774 cells, which might be a critical factor in the pathogenesis of MAYV [25]. Influenza virus invasion might lead to oxidative stress, which ultimately results in tissue damage, an inflammatory response and cell apoptosis [26]. Furthermore, it has been reported that redox alterations are crucial factors in aspects of HIV-1 pathogenicity such as neurotoxicity and dementia, exhaustion of $CD4^+$/ $CD8^+$ T-cells, predisposition to lung infections, and so on [27]. Therefore, it can be speculated that the changes in the level of oxidation induced by CV-A16 infection might not only alter the host's immune system but might also be involved in neurotoxicity caused by CV-A16 infection. In the KEGG pathway analysis, the enriched "Gonadotropin-releasing hormone receptor pathway" included 4 genes, FOSB, COX2, FOS and ID2. Previous studies have demonstrated that the gonadotropin-releasing hormone receptor pathway plays pivotal roles in the

**Table 3. Gonadotropin-releasing hormone receptor pathway-associated differentially expressed genes in KEGG pathway analysis.**

| Pathway | Gene Symbol | Gene Name | PANTHER Protein Class | CV-A16-12h | | CV-A16-24h | |
|---|---|---|---|---|---|---|---|
| | | | | Fold change | Log2FoldChange | Fold change | Log2FoldChange |
| Gonadotropin-releasing hormone receptor pathway | FOSB | Protein fosB | basic leucine zipper transcription factor | 0.176684489 | -2.500752704 | 0.105148361 | -3.249501739 |
| | COX2 | Cytochrome c oxidase subunit 2 | oxidoreductase | 0.29029806 | -1.784393164 | 2.45801905 | 1.297496097 |
| | FOS | Proto-oncogene c-Fos | basic leucine zipper transcription factor | 0.183157016 | -2.448847131 | 0.135560318 | -2.882993165 |
| | ID2 | DNA-binding protein inhibitor ID-2 | transcription factor | 2.243445451 | 1.165716106 | 2.567083091 | 1.360129994 |

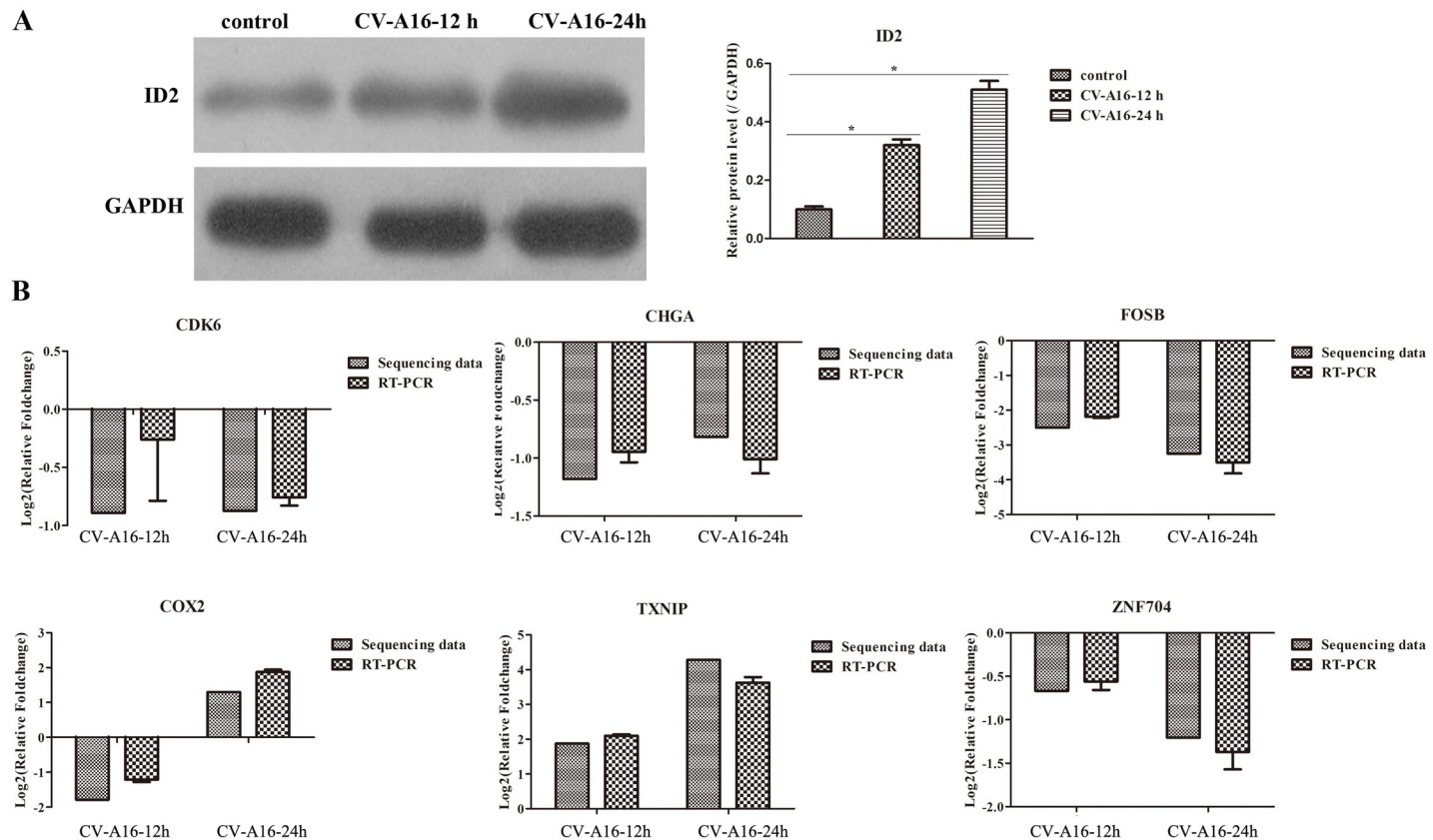

**Fig 5.** (A) The protein levels of ID2 were checked by WB in SH-SY5Y cells with CV-A16 infection. (B) A total of 6 differentially expressed genes were randomly selected for qRT-PCR validation.

control of reproductive functions in all vertebrate species [28]. However, the role of abnormalities in this pathway in CV-A16 infection is unknown. Since this study was performed on SH-SY5Y cells, ID2 in this pathway was selected for discussion, mainly because ID2 is

**Table 4. Differentially expressed non-protein-coding genes simultaneously appeared in CV-A16-12h and CV-A16-24h.**

| Gene ID | CV-A16-12h | CV-A16-24h | Type |
|---|---|---|---|
| TRNT | 0.647639572 | 0.596567542 | tRNA |
| SNORA67 | 0.528337546 | 0.606063467 | snoRNA |
| SNORD54 | 0.540954562 | 0.542743403 | snoRNA |
| MTATP6P1 | 0.398773097 | 1.510364725 | pseudo |
| MTND2P28 | 0.267285808 | 1.798882184 | pseudo |
| RNU6-1016P | 0.592867322 | 0.632896811 | pseudo |
| RNA5SP155 | 0.273357252 | 0.189724216 | pseudo |
| MIR7705 | 1.652657527 | 0.602513304 | Precursor_miRNA |
| MIR2682 | 0.452317755 | 0.376718587 | Precursor_miRNA |
| RN7SL3 | 2.048864877 | 1.768414846 | ncRNA |
| SCARNA5 | 1.540984508 | 1.678038224 | ncRNA |
| MIR675 | 0.656693615 | 0.438646659 | miRNA |
| MIR4705 | 0.40131171 | 0.479692787 | miRNA |
| MIR8063 | 0.586454676 | 0.477924334 | miRNA |

considered to be a crucial molecule that might directly influence the differentiation and development of neurons [29]. Moreover, the latest report stated that sustained high expression of ID2 could also cause damage to the nervous system [30]; thereby ID2 actually plays the role of "a double-edged sword" in the nervous system. Additionally, the WB results clearly showed that ID2 was significantly up-regulated in CV-A16-infected SH-SY5Y cells. Thus, it was concluded that changes in ID2 might participate in neurological symptoms caused by CV-A16 infection.

Additionally, the human genome sequencing project has definitely uncovered that in the human whole genome, almost 98% of the genome is dynamically, pervasively and actively transcribed into non-coding RNAs (ncRNAs), which were formerly regarded as transcriptional "noise" or body "garbage" [31]. Nevertheless, in the recent years, numerous literatures have convincingly exhibited that ncRNAs participate in controlling every level of gene expression in diverse cellular processes, and the dysregulated expression of ncRNAs strongly contributes to the initiation and progression of various diseases, including infectious diseases [32]. Moreover, it has been reported that CV-A16 infection could induce abnormal ncRNA expressions. For instance, CV-A16 might penetrate the blood-brain barrier and then enter the CNS by downregulating miR-1303, which disrupts junctional complexes by directly regulating MMP9 and ultimately causing pathological CNS changes [33]. Hence, in the current study, we also paid attention to the abnormal changes in ncRNA induced by CV-A16 infection. The results presented that a total of 14 differentially expressed genes simultaneously appeared in the CV-A16-12 h and CV-A16-24 h groups, chiefly including tRNA, snoRNA, pseudo, Precursor_miRNA, miRNA, etc.

In summary, this study provides a preliminary landscape of the transcriptome profiles of SH-SY5Y cells infected with CV-A16. These data pave the way for future studies of the molecular mechanisms underlying altered neurological symptoms induced by CV-A16. However, here we must declare that all *in vitro* cell model experiments cannot fully reflect a state *in vivo*. This is a common limitation of all *in vitro* cell model experiments. Thus, the *in vitro* cell model in the present study only provides a possible research direction for our future research.

## Supporting information

**S1 Fig. Viral growth curves and the CPE of SH-SY5Y cells with CV-A16 infection.** (A) The replication kinetics of CV-A16. (B) CPE of SH-SY5Y cells (200×amplication).
(TIF)

**S2 Fig. Quality evaluation of transcriptome sequencing data.** (A) QC results. (B) GC content.
(TIF)

**S3 Fig.** (A) Gene structure of dysregulated differentially expressed genes. (B) Chromosomal distribution of dysregulated differentially expressed genes.
(TIF)

**S1 Table. Relevant information of gene and primer sequences for strand-specific qRT-PCR.**
(DOCX)

**S2 Table. The number of up-regulated and down-regulated genes in different gene types.**
(DOCX)

**S1 File.**
(PDF)

## Author Contributions

**Conceptualization:** Yajie Hu, Jie Song, Yunhui Zhang.

**Data curation:** Zhen Yang.

**Funding acquisition:** Jie Song.

**Investigation:** Danxiong Sun, Mingmei Zhong.

**Methodology:** Shenglan Wang.

**Software:** Zhen Yang, Danxiong Sun.

**Supervision:** Mingmei Zhong, Mudong Wen.

**Validation:** Mudong Wen.

**Writing – original draft:** Yajie Hu.

**Writing – review & editing:** Jie Song, Yunhui Zhang.

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
