## [Editor Report · Decision Letter 0]

18 May 2020

PONE-D-20-10158

Comprehensive analysis of differential expression profiles via transcriptome sequencing in SH-SY5Y cells infected with CA16.

PLOS ONE

Dear Professors Song and Zhang,

Thank you for submitting your manuscript to PLOS ONE. After careful consideration, we feel that it has merit but does not fully meet PLOS ONE’s publication criteria as it currently stands. Therefore, we invite you to submit a revised version of the manuscript that addresses the points raised during the review process.

It would be important to provide a more clear rational for the use of the neuroblastoma cell line SH-SY5Y for these studies, as well as provide a more elaborated discussion of the limitations of the cell based experimental model used in the paper. Likewise, it would be important, if possible, to incorporate information about how changes in gene expression identified by the RNAseq correlate with protein expression levels.

We would appreciate receiving your revised manuscript by Jul 02 2020 11:59PM. To enhance the reproducibility of your results, we recommend that if applicable you deposit your laboratory protocols in protocols.io, where a protocol can be assigned its own identifier (DOI) such that it can be cited independently in the future. For instructions see: http://journals.plos.org/plosone/s/submission-guidelines#loc-laboratory-protocols

We look forward to receiving your revised manuscript.

Kind regards,

Juan C. de la Torre, Ph.D.

Academic Editor

PLOS ONE

Journal Requirements:

2. We note that you are reporting an analysis of a microarray, next-generation sequencing, or deep sequencing data set. PLOS requires that authors comply with field-specific standards for preparation, recording, and deposition of data in repositories appropriate to their field. Please upload these data to a stable, public repository (such as ArrayExpress, Gene Expression Omnibus (GEO), DNA Data Bank of Japan (DDBJ), NCBI GenBank, NCBI Sequence Read Archive, or EMBL Nucleotide Sequence Database (ENA)). In your revised cover letter, please provide the relevant accession numbers that may be used to access these data. For a full list of recommended repositories, see http://journals.plos.org/plosone/s/data-availability#loc-omics or http://journals.plos.org/plosone/s/data-availability#loc-sequencing.

Additional Editor Comments (if provided):

This paper by Hu and colleagues examines changes in the transcriptome of the human neuroblastoma cell line SH-SY5Y following infection with coxsackievirus A16 (CV-A16), one of the causative agents of hand-foot-and-mouth disease (HFMD) in children, but that has been also implicated with the development of severed neurological complications that can result in death.

The molecular bases whereby CV-A16 causes neurological symptoms are little understood, hence the significance of studies aimed at examining the impact of CV-A16 infection on the neuronal gene expression program. To this end the authors have used infection of SH-SY5Y cells as cell-based model.

The overall experimental design is straight forward and the authors have used appropriate RNAseq and bioinformatics tools to analyze changes in mRNA levels in CV-A16 infected SH-SY5Y, and use qRT-PCR on a group of selected genes to validate the results from RNAseq.

The main weakness of the paper is the lack of functional studies examining how the identified changes in host cellular gene expression in CV-A16 infected SH-SY5Y may contribute to neurological symptoms. Moreover, all the data presented relates to changes in RNA levels without showing whether these changes correlated with changes at the protein level, which would be the relevant ones in terms of mechanisms of neuropathology.

Another issue that needs additional discussion relates to the limitations of using a neuroblastoma cell line to examine the potential effects of CV-A16 on neuronal gene expression and associated neurological disturbances. It seems that the use of human iPSC derived neurons could provide a more relevant cell system, more so if glia cells are included in the experimental model of infection.

Figure S1 shows that CV-A16 induces a robust cytopathic effect at 24 hours post-infection and therefore many of the transcriptional changes observed may reflect changes associated with cellular death, and many of them might not be specific to CV-A16. It would be important to compare transcriptome changes caused by CV-A16 with those triggered by a non-viral cytotoxic stimulus.
---

## [Author Response · Author response to Decision Letter 0]

18 Jun 2020

Dear Editors and Reviewers:

Thank you for your comments and suggestions concerning our manuscript entitled “Comprehensive analysis of differential expression profiles via transcriptome sequencing in SH-SY5Y cells infected with CV-A16” (Manuscript ID: PONE-D-20-10158). Those comments are all valuable and very helpful for revising and improving our paper, as well as the important guiding significance to our researches. We hope our revised manuscript meet with your approval.

In addition, we need to state here that we have followed the instructions of the magazine. Firstly, we logged into our account, located the manuscript record, and then checked for the action link “View Attachments”, but there were no attachment files to be viewed (See below). Therefore, we think there may be no reviewer’s comments, and there is no response to the reviewers’ comments in this reply.

Yours sincerely,

Yunhui Zhang, Corresponding author

Department of Respiratory Medicine, The First People’s Hospital of Yunnan province, Kunming, 650022, China.

E-mail: zhangyh123kh@163.com

Response to editor’s comments

Journal Requirements:

Reply: Thank you for your reminder. We will strictly follow the PLOS ONE’s style requirements.

2. We note that you are reporting an analysis of a microarray, next-generation sequencing, or deep sequencing data set. PLOS requires that authors comply with field-specific standards for preparation, recording, and deposition of data in repositories appropriate to their field. Please upload these data to a stable, public repository (such as ArrayExpress, Gene Expression Omnibus (GEO), DNA Data Bank of Japan (DDBJ), NCBI GenBank, NCBI Sequence Read Archive, or EMBL Nucleotide Sequence Database (ENA)). In your revised cover letter, please provide the relevant accession numbers that may be used to access these data. For a full list of recommended repositories, see http://journals.plos.org/plosone/s/data-availability#loc-omics or http://journals.plos.org/plosone/s/data-availability#loc-sequencing.

Reply: Sorry for this negligence. The sequencing data have been submitted to the Gene Expression Omnibus (GEO) database (www.ncbi.nlm.nih.gov/geo/) under the accession number GSM4593022 (Control group), GSM4593023 (CV-A16-12 h group) and GSM4593024 (CV-A16-24 h group).

Additional Editor Comments (if provided):

This paper by Hu and colleagues examines changes in the transcriptome of the human neuroblastoma cell line SH-SY5Y following infection with coxsackievirus A16 (CV-A16), one of the causative agents of hand-foot-and-mouth disease (HFMD) in children, but that has been also implicated with the development of severed neurological complications that can result in death.

The molecular bases whereby CV-A16 causes neurological symptoms are little understood, hence the significance of studies aimed at examining the impact of CV-A16 infection on the neuronal gene expression program. To this end the authors have used infection of SH-SY5Y cells as cell-based model.

The overall experimental design is straight forward and the authors have used appropriate RNAseq and bioinformatics tools to analyze changes in mRNA levels in CV-A16 infected SH-SY5Y, and use qRT-PCR on a group of selected genes to validate the results from RNAseq.

1. The main weakness of the paper is the lack of functional studies examining how the identified changes in host cellular gene expression in CV-A16 infected SH-SY5Y may contribute to neurological symptoms. Moreover, all the data presented relates to changes in RNA levels without showing whether these changes correlated with changes at the protein level, which would be the relevant ones in terms of mechanisms of neuropathology.

Reply：Thank you very much for your comments. In the transcriptome sequencing analysis of this article, it did not directly indicate which differentially expressed genes induced by CV-A16 infection may be the cause of neurological symptoms. But it can be concluded from the analysis of transcriptome sequencing that CV-A16 infection-induced changes in oxidation levels might be directly participated in the pathogenesis of CV-A16 infection. Moreover, among these differentially expressed genes in enriched “Gonadotropin-releasing hormone receptor pathway”, ID2 is not only associated with oxidative stress, but also with nervous system development. Early research showed that ID2 can promote the development of the nervous system1, and the latest research found that sustained high expression of ID2 can cause damage to the nervous system2. Therefore, the effect of ID2 on brain tissue is hailed as a “double-edged sword” and we also selected it to examine its protein levels. It was found that after CV-A16 infection, ID2 showed high expression in SH-SY5Y cells (Fig. 5A).

2. Another issue that needs additional discussion relates to the limitations of using a neuroblastoma cell line to examine the potential effects of CV-A16 on neuronal gene expression and associated neurological disturbances. It seems that the use of human iPSC derived neurons could provide a more relevant cell system, more so if glia cells are included in the experimental model of infection.

Reply：Thank you for your constructive advice. Actually, in many studies of neurotropic viruses, SH-SY5Y cells are often used as susceptible cells to explore their pathogenic mechanism in vitro. For instance, docosahexaenoic acid (DHA), an omega-3 polyunsaturated fatty acid, has a potential anti-inflammatory and neuroprotective effect against ZIKV infection in these neuron-like cells (these were SH-SY5Y cells)3; upregulation of antioxidants including SESN2 and, also, the xCT antiporter occurs to counteract the oxidative stress elicited by JEV infection in SH-SY5Y neuroblastoma cells4. Moreover, in a large number of previous studies, EV-A71, which is also the main pathogen of HFMD, also adopted SH-SY5Y cells to explore its potential neuropathic mechanism. For example, EV-A71 induced apoptosis of SH‑SY5Y cells through stimulation of endogenous let-7b expression5. Therefore, in this study, in order to explore the transcriptome changes after CV-A16 infected SH-SY5Y cells, we also selected SH-SY5Y cells as a susceptible nerve cell for research.

3. Figure S1 shows that CV-A16 induces a robust cytopathic effect at 24 hours post-infection and therefore many of the transcriptional changes observed may reflect changes associated with cellular death, and many of them might not be specific to CV-A16. It would be important to compare transcriptome changes caused by CV-A16 with those triggered by a non-viral cytotoxic stimulus.

Reply：Thank you for your suggestion. The effect of inducing cell death after viral infection is indeed not specific to the CV-A16 virus. Many viral infections will trigger cell death. In this paper, after analysis of transcriptome sequencing, the results showed that changes in oxidative stress may play an important role in the pathogenesis of CV-A16. Moreover, abnormal changes in ncRNAs may also be involved in the pathogenesis of CV-A16. Although CV-A16 infection triggered the formation of cell death, it was not clearly indicated in the transcriptome sequencing analysis. Therefore, in this article, we did not focus on the situation of virus infection-induced cell death.

---

## [Editor Report · Decision Letter 1]

22 Jun 2020

PONE-D-20-10158R1

Comprehensive analysis of differential expression profiles via transcriptome sequencing in SH-SY5Y cells infected with CA16 .

PLOS ONE

Dear Dr. Zhang,

Thank you for submitting your manuscript to PLOS ONE. After careful consideration, we feel that it has merit but does not fully meet PLOS ONE’s publication criteria as it currently stands. Therefore, we invite you to submit a revised version of the manuscript that addresses the points raised during the review process.

I concur with the comment made by the reviewer indicating that the discussion section of the paper should present a brief discussion about the limitations of using a neroblastoma cell line to infer the outcome of CV-16 brain infection at the whole organism level.

We look forward to receiving your revised manuscript.

Kind regards,

Juan Carlos de la Torre, Ph.D.

Academic Editor

PLOS ONE

Additional Editor Comments (if provided):

The discussion section of the paper should incorporate a brief discussion about the limitations of using a neuroblastoma cell line to infer viral induced changes in neuronal gene expression in the context of a whole organism.

---

## [Editor Report · Decision Letter 2]

6 Jul 2020

PONE-D-20-10158R2

Comprehensive analysis of differential expression profiles via transcriptome sequencing in SH-SY5Y cells infected with CV-A16

PLOS ONE

Dear Dr. Zhang

In the previous revision of your paper, I indicated that before the paper being considered suitable for publication in PLOS ONE it was necessary to incorporate into the discussion section of the paper a brief discussion about the limitations of using a neuroblastoma cell line to infer viral induced changes in neuronal gene expression in the context of a whole organism.

However your R2 version of the paper has ignored the recommendation.

Before I can proceed with the acceptance of the paper for publication, this issue needs to be addressed.

We look forward to receiving your revised manuscript.

Kind regards,

Juan Carlos de la Torre, Ph.D.

Academic Editor

PLOS ONE

Additional Editor Comments (if provided):

Dear Dr. Zhang

In the previous revision of your paper, I indicated that before the paper being considered suitable for publication in PLOS ONE it was necessary to incorporate into the discussion section of the paper a brief discussion about the limitations of using a neuroblastoma cell line to infer viral induced changes in neuronal gene expression in the context of a whole organism.

However your R2 version of the paper has ignored the recommendation.

Before I can proceed with the acceptance of the paper for publication, this issue needs to be addressed.

---

## [Decision Letter · Decision Letter 3]

12 Oct 2020

Comprehensive analysis of differential expression profiles via transcriptome sequencing in SH-SY5Y cells infected with CV-A16

PONE-D-20-10158R3

Dear Dr. Zhang,

We’re pleased to inform you that your manuscript has been judged scientifically suitable for publication and will be formally accepted for publication once it meets all outstanding technical requirements.

Kind regards,

Juan Carlos de la Torre, Ph.D.

Academic Editor

PLOS ONE

Additional Editor Comments (optional):

The authors have addressed adequately the minor comments raised about their revised paper.

There are not additional concerns regarding the scientific content of this revised version of the paper.

Reviewers' comments:

Reviewer's Responses to Questions

**Comments to the Author**

1. If the authors have adequately addressed your comments raised in a previous round of review and you feel that this manuscript is now acceptable for publication, you may indicate that here to bypass the “Comments to the Author” section, enter your conflict of interest statement in the “Confidential to Editor” section, and submit your "Accept" recommendation.

Reviewer #1: All comments have been addressed

2. Is the manuscript technically sound, and do the data support the conclusions?

Reviewer #1: Yes

3. Has the statistical analysis been performed appropriately and rigorously? 

Reviewer #1: Yes

4. Have the authors made all data underlying the findings in their manuscript fully available?

Reviewer #1: Yes

5. Is the manuscript presented in an intelligible fashion and written in standard English?

Reviewer #1: Yes

6. Review Comments to the Author

Reviewer #1: The authors utilized transcriptome sequencing technology to identify mechanisms of coxsackievirus A16 (CV-A16) pathology in SH-SY5Y cells. CV-A16 is associated with childhood illnesses such as hand, foot, and mouth disease and central nervous system pathology. Utilizing this approach, the authors demonstrated changes in oxidation levels and the “gonadotropin-releasing hormone receptor pathway” by GO term analysis or KEGG pathway analysis, respectively. qRT-PCR was utilized to verify the transcriptome sequencing data for some genes. SH-SY5Y cells have been previously utilized as a neuronal model to study enterovirus (and other virus) replication and cytopathic effects on the host cell. These investigators have previously published similar studies on enterovirus A71 (EV71) infection of SH-SY5Y cells (Hu et al, Virus Res. 2020). The previous studies also revealed “enrichment” of “gonadotropin-releasing hormone receptor pathway” genes, but also “CCKR signaling” in SH-SY5Y cells infected with EV71. The new study provides useful information regarding the transcriptome profile for SH-SY5Y cells following CV-A16 infection.

Specific Comments and suggested improvements:

The authors utilize SH-SY5Y transformed cells originally isolated from a patient with a neuroblastoma. Although these transformed cells have been utilized historically for in vitro models of neuronal function and differentiation, far better and more relevant primary neural stem cell culture models (iPSC or hESCs) currently exist to determine virus-mediated effects on neuronal cell function and differentiation.

The authors utilize western blot to verify upregulation of ID2 protein expression, which they describe as a crucial molecule influencing neuronal differentiation. However, protein levels for other genes were not inspected. Also, no examination of ID2 over-expression following infection and downstream effects on related neuronal development genes was ascertained.

The study would benefit with in vivo, or alternatively primary cell culture analysis of changes of expression levels of genes following CV-A16 infection.

The authors speculate as to how changes in oxidation levels or the ‘gonadotropin-releasing hormone receptor pathway” might participate in the pathogenesis of CV-A16 infection. However, no experiments were included to validate such speculation. Also, the central nervous system is composed of many other cell types. Examining expression changes following CV-A16 infection in a single transformed cell type is unlikely to clarify the mechanism of virus-induced neuropathy.

7. PLOS authors have the option to publish the peer review history of their article (what does this mean?). If published, this will include your full peer review and any attached files.

Reviewer #1: No

---

## [Editor Report · Acceptance letter]

28 Oct 2020

PONE-D-20-10158R3 

Comprehensive analysis of differential expression profiles via transcriptome sequencing in SH-SY5Y cells infected with CV-A16  

Dear Dr. Zhang:

I'm pleased to inform you that your manuscript has been deemed suitable for publication in PLOS ONE. Congratulations! Your manuscript is now with our production department. 

Kind regards, 

on behalf of

Dr. Juan Carlos de la Torre 

Academic Editor

PLOS ONE